# Determining the Beginning of Potato Tuberization Period Using Plant Height Detected by Drone for Irrigation Purposes

Sarah Martins [1,*] , Rachid Lhissou [1] , Karem Chokmani [1] and Athyna Cambouris [2]

1 Institut National de la Recherche Scientifique—INRS, Québec, QC G1K 9A9, Canada
2 Agriculture Agroalimentaire Canada—AAC, Québec, QC G1V 2J3, Canada
* Correspondence: sarahca.martins@gmail.com

**Abstract:** Insolation and precipitation instability associated with climate change affects plant development patterns and water demand. The potato root system and soil properties lead to water vulnerability, impacting crop yield. Regarding potato physiology, plants stop growing when the root depth stabilizes, and then the tuberization period begins. Since this moment, water supply is required. Consequently, an approach based on plant physiology may enable farmers to detect the beginning of the irrigation period precisely. Remote sensing is a fast and precise method for obtaining surface information using non-invasive data collection. The database comprises root depth (RD) and plant height (H) data collected during 2019, 2020, and 2021. This research aims to develop a dynamic approach based on remote sensing and crop physiology to accurately determine the beginning of the tuberization period, called here the irrigation critical point (ICP). The results indicate a high correlation between RD and H (>0.85) which is independent of in-field soil and relief variations > 0.95). Further, plant growth rate corroborates the correlation results with decreasing patterns in time ($R^2 > 0.80$), independent of environmental variations. In short, it was possible to determine the ICP based on the crop growth dynamics, independently of climate variations, field placement, or irrigation system.

**Keywords:** potato physiology; remote sensing; smart farming; UAV; water supply

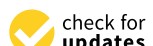



## 1. Introduction

Irrigated crop fields are responsible for almost 40% of global food production, accounting for about 20% of all crop fields worldwide [1,2]. Furthermore, agriculture is responsible for 80% of all water consumption on a planetary scale, most of which (2/3) is used for irrigating fields [1,3]. Climate change is a critical issue for agriculture, not only because of resource depletion (water and land growing demands) or pollution (chemical lixiviation), but also regarding plant development pattern adjustments and water demand across growing seasons [1,4]. Climate change increases the frequency of extreme meteorological events, and it modifies the pattern of precipitation and insolation [1,5–9]. Similarly, we have noticed through observations in the field that crop growth dynamics are changing as a natural accommodation to new weather patterns [8,10–12]. Moreover, the seed/plantation schedule is not a reliable tool anymore for determining when farmers may start irrigation, because biological standards are undoubtedly changing [10].

According to the Food and Agriculture Organization of the United Nations (FAO), the potato is one of the most important crop globally, and its consumption has been increasing by almost 5% per year since 1998, especially in developing countries [13]. Using fully adequate practices, which may include nutrients and irrigation when necessary, potato crop fields can produce 25 to 35 tons/ha after a 120-day growing season in temperate climates [14–16]. For every cubic meter of water, potatoes can provide 5600 kcal of dietary energy, 150 g of protein, and many micronutrients [14,17]. Additionally, the proportion of chronically undernourished people is growing worldwide [17].

Despite the potato's high yield rates and nutritional characteristics, it is considered the "food of the future", because this crop, unlike grain foods, is not marketed as a commodity [17]. Some advantages related to potato cultivation are: (i) the crop field is relatively easy to manage; (ii) it is a relatively low-cost implementation culture; and (iii) the potato is a high-nutrition plant in comparison with other tubers or cereals [14–17]. These characteristics place potatoes in the FAO's list of recommended crops to ensure food security with highly nutritional crops protecting natural resources [14,16,17].

Within mild climates, potato is considered a crop with high water demand because of soil characteristics and plant physiology. Generally, potato is cultivated in well-drained soils, like sandy soil, to avoid diseases and to reduce soil resistance, benefiting tuber development [18]. According to many authors [14,15,19], the primary characteristics of potato physiology are: (i) tuber maturity and quality depend on water and nutrients; (ii) potatoes have a shallow root system; (iii) plants stop growing when the root achieves its maximum root depth, and then plant energy is consecrated to tuber development—the beginning of the tuberization period [14,15,19–22].

The first characteristic indicates the environmental factors for a healthy crop field [10,14,15,23]. The second is a major physiological constraint to the irrigation schedule because it makes this crop vulnerable to water stress [5,6,15]. The last one can be used to reduce the vulnerability of potato to water supply [11,12,20,21,24,25]. As this study has is in the application framework of irrigation vulnerability, we are focusing mainly on the last characteristic.

According to many authors [14,20–22,24,26], the photosynthesis product (dry matter) is the basis of the relationship between plant growth, root depth development and the beginning of tuberization. In short, the dry matter distribution changes over potato plant phenological phases, from the foliage at the beginning of the plant growth cycle to the tubers as soon as the tuberization phase begins, favorizing tuber development instead of plant height [14,20,21]. Additionally, potato plants have circadian genes affecting the expression of clock-regulating specific genes [20–22,24,27]. These genes are active in photoperiod-dependent tuber formation, indicating that environmental factors greatly affect tuberization triggers and growth hormone inhibitors [20–22,24,27]. Both meteorological conditions and the tuberization phase may be considered for crop yield improvement [20–22,24,27].

In situ data are often associated with punctual and invasive sampling (e.g., plant extraction or taking soil samples). In contrast, remote sensing is a non-destructive approach for massive data acquisition with accurate spatial reference [28]. As they are acquired directly from the surface, in processing remotely sensed data generalization methods commonly associated with punctual data collection should be avoided [2,28–30]. In the same way, it is possible to reduce the costs related to staff, time consumption, and laboratory analysis by using remote sensing platforms for data acquisition [28].

Remote sensing data can be acquired by different sensors such as satellites, aircraft, and drones, also known as Unmanned Aerial Vehicles (UAVs) [28,31]. The latter are the most flexible because it is possible to change the on-board captor without changing the UAV unit [3,29,31–33]. This means that drones allow us to play with data parameters such as image resolution, specifically spatial, spectral, temporal, and radiometric resolutions [31–36]. It is possible to apply photogrammetry techniques to drone images to produce a field surface Digital Elevation Model—DEM [37,38]. Such data are suitable for accessing plant height evolution over time.

Since well-adapted spatial resolution can accurately detect crop height increase and temporal resolution can be conveniently set according to plant growth dynamics, remote sensing and photogrammetry should be useful in obtaining the growth curve of potato crops throughout the growing season. By tracking plant height (H), it may be possible to detect the beginning of the tuberization period through the relationship between the stabilization moment of plant height and root depth [14,37–39].

In this context, the challenge regarding this research is to accurately establish the beginning of the tuberization period despite the influence of climate variations on potato

crop growth development. This moment is here called the irrigation critical point (ICP) because the water supply is a concern for the farmer from the beginning of the tuber development phase. The ICP should be interpreted as the checking point in the irrigation schedule since plants still need to be observed during the growing season. In addition, to better supply the crop fields, other techniques such as soil sampling are available to determine the retention curve and better estimate irrigation amounts.

Therefore, knowing precisely when the water supply is needed should prevent yield damages related to irrigation schedule errors (early or late irrigation [10–12,15,26,27]. In this way, using the crop growth dynamics acquired by photogrammetry to establish when the tuberization period starts may decrease costs (staff and laboratory), and improve the time-consuming analysis step. Similarly, this method provides surface information without invasive sampling or generalizations [28].

As a hypothesis, the ICP can be used as an indicator of the beginning of the tuberization period, depending on plant height or root depth. Since the ICP is delivered by plant growth dynamics, it can be determined by remotely sensed photogrammetry without using in situ punctual data or even generalization methods.

Based on this challenge, this study aims to develop an approach based exclusively on the potato crop development dynamics to determine the ICP. The innovation presented here is to replace the irrigation decision based on empirical/visual determination or the seeding/planting calendar with an approach exclusively based on the dynamics of plant development itself. The goal of using plant growth dynamics is related to the fact that plants are adapting their growth development because of climate change [8,27,40]. This means that the ICP may vary over time and field placement, as was observed in this study.

This plant-based approach has an important impact on both economics and crop science. The former refers to improving crop yield, since irrigating at the right moment may reduce the risks related to irrigating early or late, as mentioned. Further, the costs associated with irrigation structure and water consumption may decrease [27]. Secondly, this represents an innovative, flexible, and dynamic manner for dealing with irrigation, which may be an inspiring approach for either other tuber plants or different crops.

It is important to highlight that this project is a part of a major ongoing project. Therefore, the approach to deliver the ICP, as it is proposed here, will be used as a trigger for start an automated application based on artificial intelligence to determine irrigation schedules based on plant evapotranspiration rates as well as meteorological information, in real-time.

## 2. Materials and Methods

### 2.1. Field Characterization

Data collection proceeded in three potato crop fields during three growing seasons, namely Champ 41 (2019—4 hectares), Champ Réal (2020—6 hectares), and Champ Doris (2021—5 hectares) all located in the Sainte-Catherine-de-la-Jacques-Cartier, an area renowned for its potato production (46°51′ N, 71°37′ W), near Québec City, QC, Canada. These crop fields had different irrigation methods and schedules, namely center (2019 and 2021) and rain-fed irrigation (2020). The potato cultivar grown in all of them was Russet Burbank. Moreover, all fields have a similar soil type and were classified as Haplorthod, well-drained sandy soil [40]; Figure 1.

The Quebec agricultural region lies in a humid and cold continental climate characterized by an annual average temperature of 2 °C and an average summer temperature over 17 °C [41]. As for precipitation standards, this region has an amount of 1000 mm per year (solid and liquid) and a liquid precipitation average of up to 100 mm in summer [41]. According to the official models for this region [9], the future climate scenarios are characterized by temperature augmentation, increased occurrence of extreme events, incensement of liquid precipitation volume, and rain occurrence becoming concentrated over the sea-

sons. Additionally, this study strongly recommends farmer adaptation regarding irrigation methods [9].

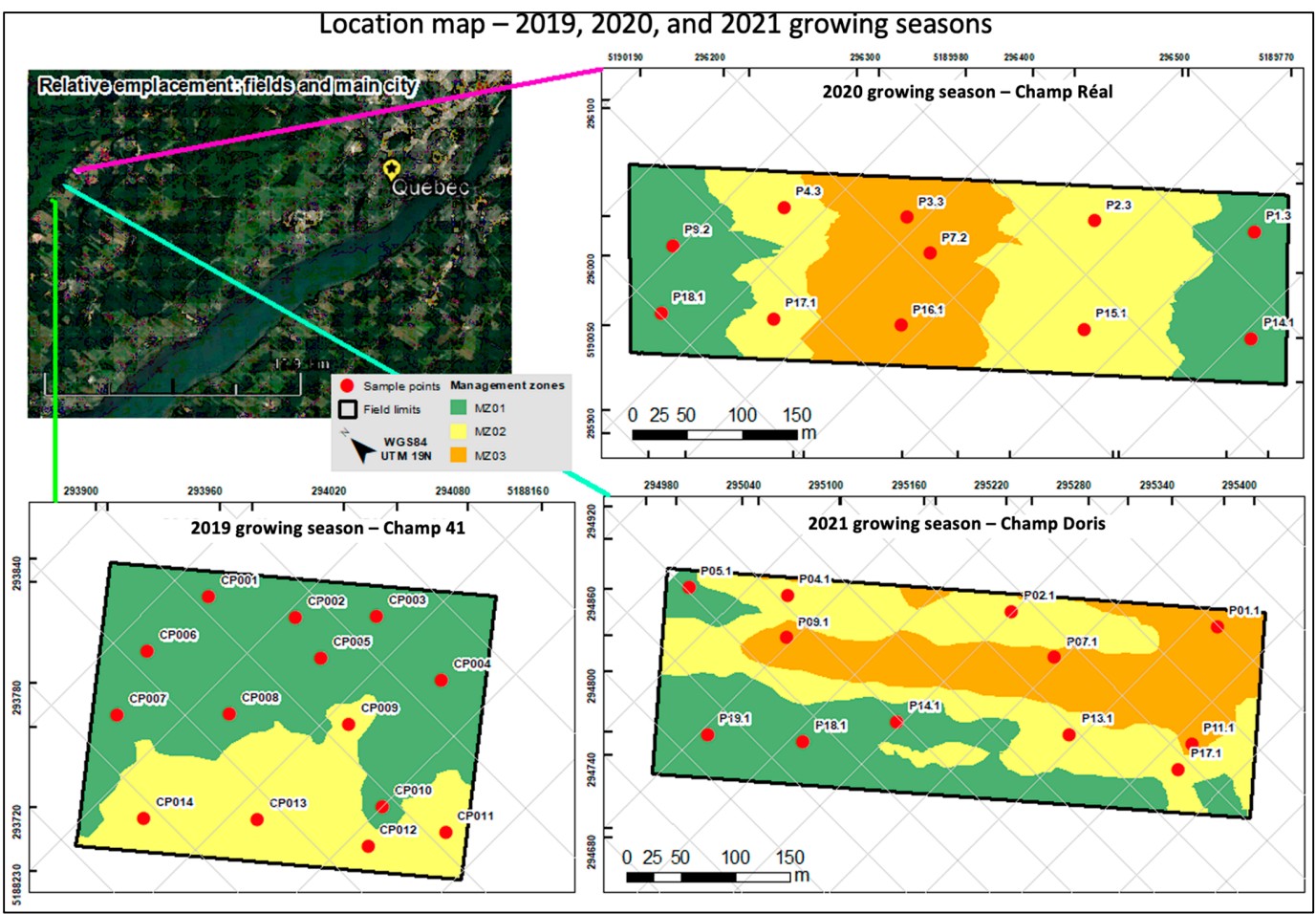

**Figure 1.** The placement of the growing season fields and the distribution of sample points and MZ configuration.

### 2.2. Data Collection and Processing

The database is composed of in situ data, as well as remote sensing photogrammetry data (Figure 2).

1.  Root depth: 2019 and 2020 growing seasons. Sample points were georeferenced at the field for taking direct measures using a precise measure tape. One plant per sample point was measured. The same plant was not measured twice over the growing season to avoid bias due to eventual damage in roots and soil manipulation.
2.  Plant height: 2021 and 2021 growing seasons. Plant height was obtained using a MicaSens RedEdge (DJI—China) camera boarded on a Matrix-M200 drone (DJI—China). The acquisition frequency and period are summarized in Table 1. Image processing to have by-week MDEs was performed using Drone Deploy (Drone Deploy—USA) and Pix4D (Pix4D—Swiss) software.
3.  Meteorological data: An in situ weather station (Hobo—USA) was installed in a corner of each field to measure the parameters that directly affect plant growth dynamics. The meteorological parameters used were temperature (maximum—Tmax, minimum—Tmin, and average—Tave), solar radiation (SRmax), and liquid precipitation (maximum—Pmax, quantity—Pq, and cumulated—Pc).

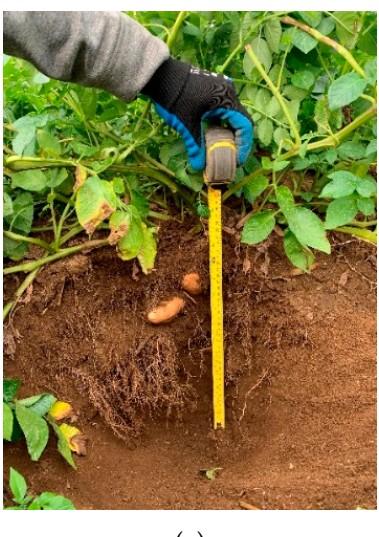

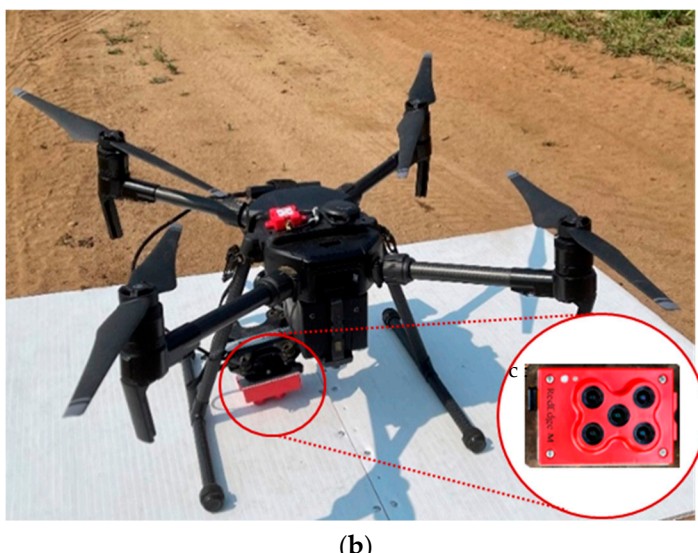

(**a**)                 (**b**)

**Figure 2.** Data collection: (**a**) in situ root depth measurements, and the remote sensing data acquisition set-up (UAV—(**b**) and camera—(**c**)).

**Table 1.** Data collection characteristics per field.

| Field | Plantation | Data Collection Period | Sample Points | Management Zones | Frequency |
|---|---|---|---|---|---|
| Champ 41 | 2019/05/09 | 2019/07 to 2019/09 | 14 | 2 | weekly |
| Champ Réal | 2020/05/20 | 2020/07 to 2020/09 | 11 | 3 | weekly |
| Champ Doris | 2021/05/04 | 2021/05 to 2021/09 | 12 | 3 | weekly |

Each field was divided into management zones (MZ), with the aim of understanding differences in crop development related to in-field variability [42,43]. MZs are handy for verifying if soil properties or relief play an essential role in tuberization beginning and if they impact RD or H development.

Two main criteria determined MZ definition: relief and soil properties (physical and chemical). The former deals with water displacement into the soil profile because of slope and elevation control, namely infiltration and runoff [19,44]. The latter deals with apparent soil electrical conductivity, which is stable in time, reflecting a soil's inherent soil properties, such as texture and drainage [19,44].

MZs were determined before planting, using an MSP3 (VERIS—USA) between 0—30 cm deep to acquire measurements of apparent soil electrical conductivity, and data processing was performed with the clustering method using the ISODATA function of ArcGIS (ESRI—USA). Topography data were taken by drone and compared to the 3 m resolution MDT provided by the Canadian government (available at https://www.donneesquebec.ca/recherche/fr/dataset/produits-derives-de-base-du-lidar, accessed on 27 January 2023). The MZ classes were defined relatively to the in-field characteristics, where MZ01 corresponds to the smallest relief slope and the lowest values of apparent soil electrical conductivity. The other classes, MZ02 and MZ03 (2020 and 2021 growth seasons), represent a statistically significant improvement in both characteristics, justifying a new class nomination.

Regarding the 2019 growing season, sample points were randomly distributed in the field using ArcMap 10 (ESRI—USA). The distribution of the 2020 and 2021 sample points was coordinated with another project, because of the COVID-19 protocols. At that time, the sample point placement rules were: (i) no less than 3 points per MZ; and (ii) every field corner should have a sample point.

Briefly, a Digital Elevation Model (DEM) provides the elevation surface above the soil delivered by photogrammetric techniques applied to remote sensing data [28,37,39]. The resolution of the resulting model is directly related to the sensor used in data acquisition and the resulting precision [28,39,45]. For this study, the flight altitude (60 m) was maintained in all data collection periods to ensure equal spatial resolution for all images (5 cm) and the time series analyses, if needed. Additionally, the manufacturer indicates 3 cm as the altitude (z-axis) precision for the camera used.

For instance, drone flights were deployed during clear-sky days (avoiding cloud influences in images), and the ideal acquisition period between 10 h and 14 h was respected [28,45]. Plant H was acquired directly from weekly DEM at the sample points. The in situ data processing and statistics were conducted in Spyder 5.1.2 (Spyder IDE—Open Source) software using the Python programming language and freely available libraries, namely Pandas, Seaborn, Numpy, Matplotlib, OS, and Scipy.

## 3. Results

### 3.1. Root Depth—2019 and 2020 Growing Seasons

Although roots stopped deepening in late July (07/25) and early August (08/01) in the 2019 and 2020 growing seasons, respectively, it took 24 days until root depth stabilized in both cases; see Figure 3. This could have happened because the weather patterns changed over the seasons, but the dynamics of plant development remained the same [8,10]. In addition, the average RD values per MZ do not indicate a significant difference, as they are mostly close to 30 cm for both years.

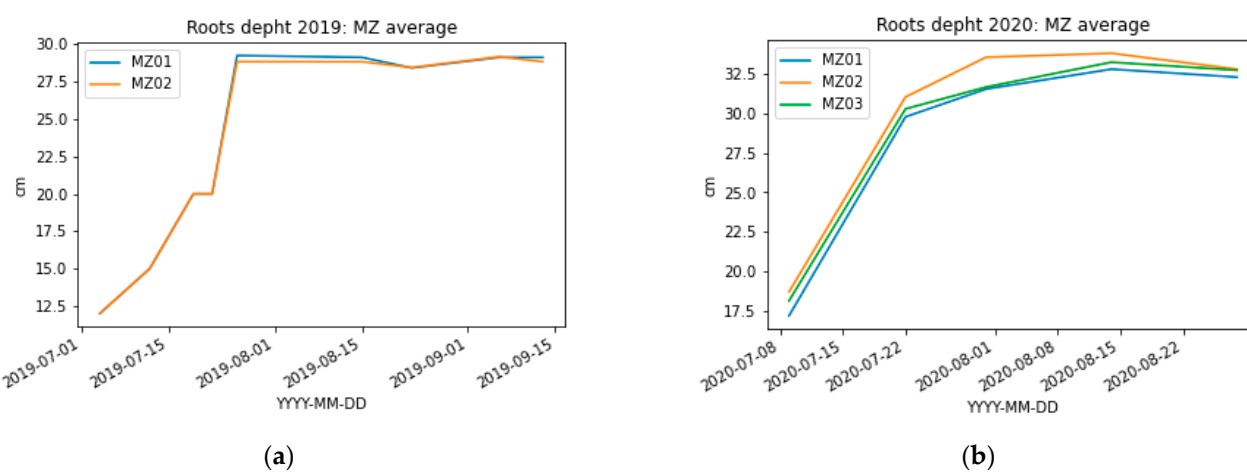

(**a**)                      (**b**)

**Figure 3.** Average of root depth evolution (in cm) by MZ obtained in 2019 (**a**) and 2020 (**b**) growing seasons.

### 3.2. Plant Height and Growth Curve—2020 and 2021 Growing Seasons

The growth curve and the respective H growth rate for both the 2020 and the 2021 growing seasons are presented in Figures 4 and 5. Regarding the H graphs, the growth curve shapes are similar, and stabilization took place in late July (2020) and early August (2021). Plants got higher in the 2020 growing season, and this is the season with the highest difference among H curves. In the 2021 growing season, all MZ had almost the same H values, probably because of the similarity in MZ properties. Further, all 2021 H values are comparable to those for MZ01 in 2020.

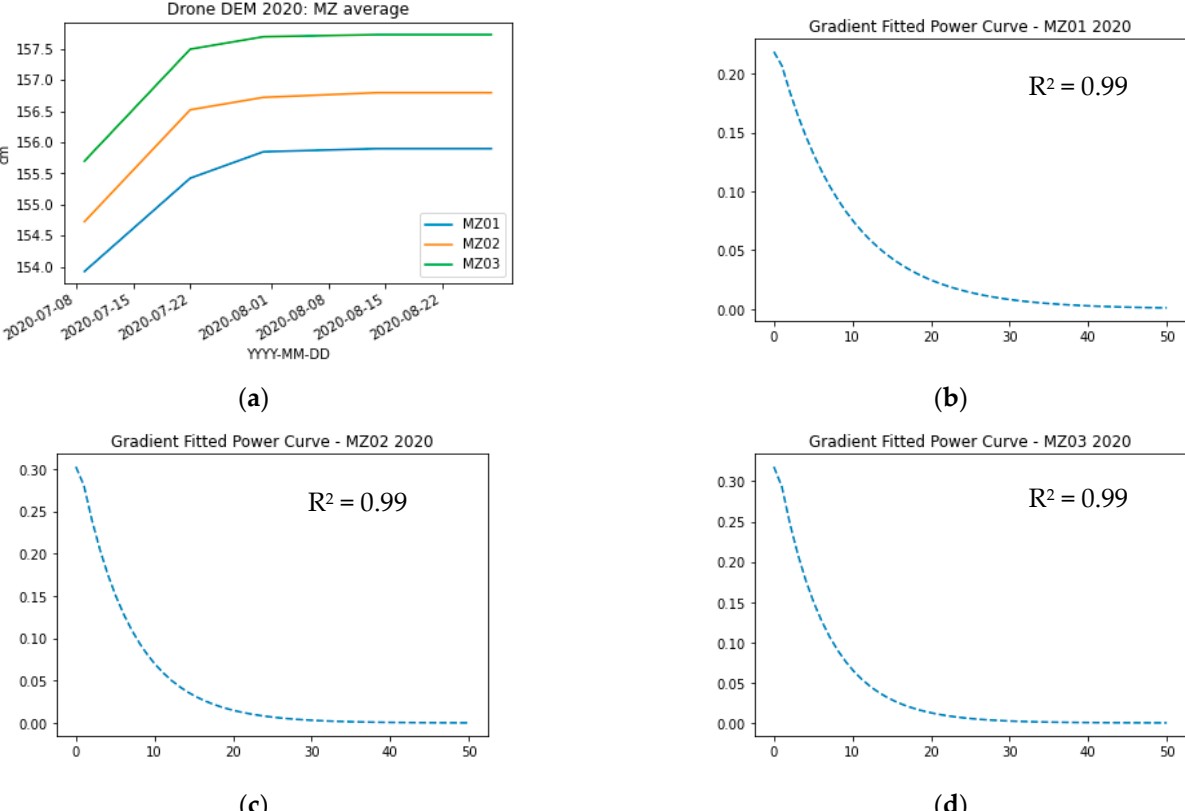

**Figure 4.** Evolution of plant height (H) over the 2020 growing season until H stabilization: Growth curve ((**a**)—cm vs date) and growth rate over management zones MZ01 (**b**), MZ02 (**c**), and MZ03 (**d**).

The H growth rate was obtained by derivation of H over time using a gradient-fitted power curve. In both cases, the H growth rate fits with the plant growth curve once the higher value is placed in the early season and the downslope is placed between days 1 and 15 (2020/07/22) or between days 1 and 60 (2021/07/10) of the 2020 and 2021 data collection periods, respectively. In both cases, the slope is well-placed in the plant height spread. Comparing the H curve shape and its related growth rate, it is possible to verify that when H stabilizes, the H rate is close to 0 (~0.05). The $R^2$ of the gradient-fitted curve was 0.99 for all 2020 MZs and above 0.98 for the 2021 MZs (MZ01 = 0.98; MZ02 = 0.99; MZ03 = 0.99).

Note that the days mentioned refer to the beginning of the season's data collection period, which means that the 15th day of the 2020 growing season occurred at the same time of the year as the 60th day of the 2021 growing season. This may have happened because the 2020 data collection started on 2020/07/09, and the 2021 data collection started on 2021/05/10. In addition, the planting day differ by 10 days from 2020 and 2021 growth seasons, the exility days are: 2020/05/15 and 2021/05/04.

### 3.3. Root Depth and Plant Height Correlation—2020 Growing Season

Plant height varied across MZ and seasons, with a range of H maximum values between 155.5 and 157.5 cm for 2020. Despite the infield variations of H values, the RD average per MZ did not vary, and the growth curve stabilization took place when the maximum values for RD were reached. Furthermore, the 2020 growing season results demonstrate that RD and H stabilized at the same time, notably 24 days before root depth stabilization (Figures 3 and 4)

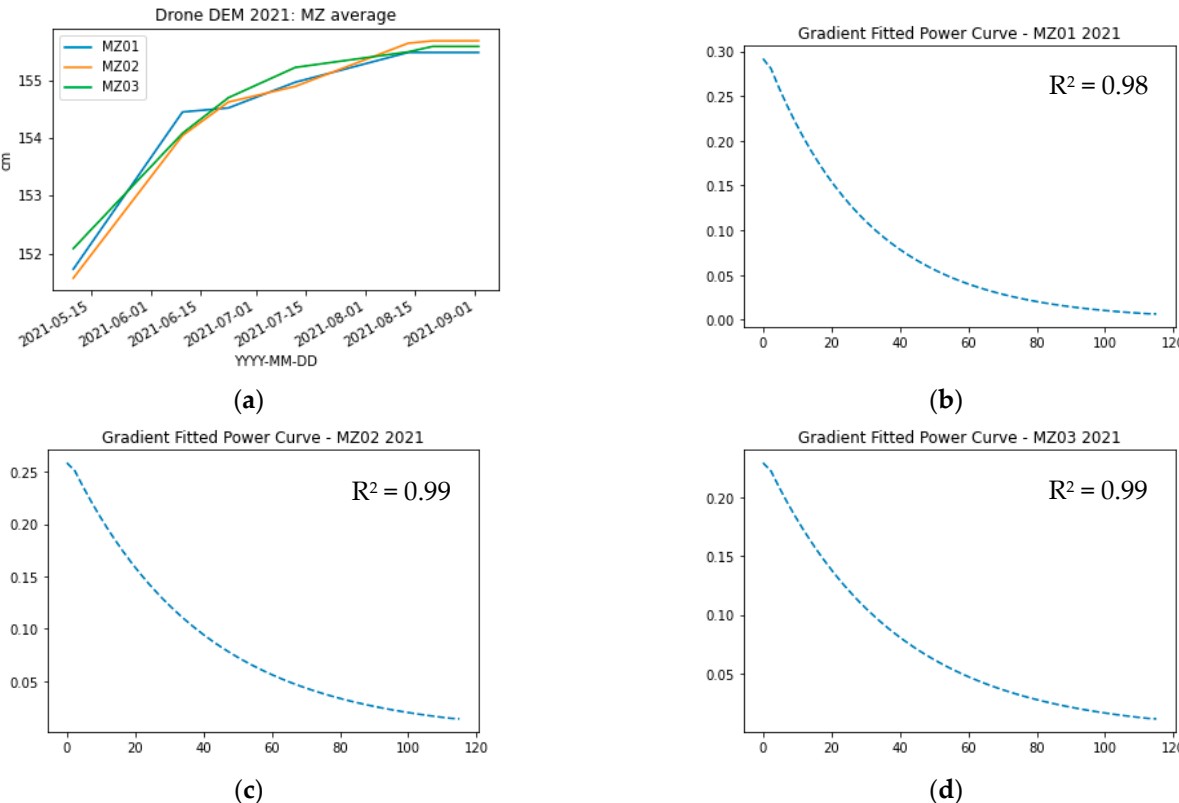

**Figure 5.** Evolution of plant height (H) over the 2021 growing season until H stabilization: Growth curve ((**a**)—cm vs date) and growth rate over management zones MZ01 (**b**), MZ02 (**c**), and MZ03 (**d**).

The investigation of Pearson correlations promoted a better understanding of the relationship between plant in-soil and out-soil behavior. Figure 6 presents the cross-correlation matrix between the average values of RD and H per MZ. The matrix shows a high correlation among both parameters, independent of the placement of sample points, with values up to 0.96.

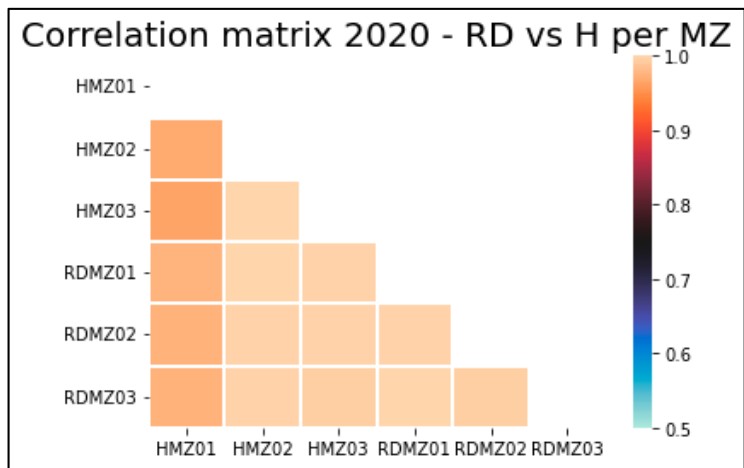

**Figure 6.** Pearson cross-correlation matrix between crop height and root depth—2020 growing season.

The correlation test of statistical significance helps to better understand the relationship between the correlated variables and their nature, while taking sample size into account [45–47]. Due to the small sample size, the t-statistic was used alongside the *p*-value with alpha = 0.05 (Table 2). Regarding the magnitude of the t-statistic values, it can be inferred that the correlation results are reliable and statistically significant, which is cor-

roborated by a *p*-value < 0.05. Moreover, the null hypothesis is rejected since the t-statistic (4–5) is above the critical value (~2). Together, the t-statistic and *p*-value support the high correlation between H and RD, ruling out the hypothesis that the correlation between the variables was a random error.

**Table 2.** Result of the statistical significance tests of the correlation between the average of RD and H per MZ over the 2020 growing season.

| Results | Test MZ01 | Test MZ02 | Test MZ03 |
|---|---|---|---|
| Var01 | HMZ01 | HMZ02 | HMZ03 |
| Var02 | RDMZ01 | RDMZ02 | RDMZ03 |
| Pearson Correlation | 0.976 | 0.996 | 0.996 |
| t-Statistic | 4.134 | 4.876 | 5.051 |
| *p*-value one-tail | 0.007 | 0.004 | 0.004 |
| t Critical one-tail | 2.132 | 2.132 | 2.132 |

This statistical significance test was proceeded using alpha = 0.05.

*3.4. Meteorological Variation during Plant Development*

The meteorological data used in this section cover the period between the planting day and either the RD stabilization for the 2019 growth season, or growth curve stabilization for 2020 and 2021 growth seasons, both representing the tuberization period beginning, named here the ICP for each growth season.

The crop development period until the RD maximum value was reached (2019) or growth curve stabilization (2020 and 2021) was variable according to growing seasons. The 2019 and 2020 growing seasons had almost 80 days, and the 2021 growing season had 98 days until stabilization. The difference in the number of days until H max is probably related to meteorological constraints and the day of planting.

Table 3 and Figure 7 present the meteorological data for the three growing seasons until the ICP achievement. The 2019 and 2020 growing seasons had almost the same features and distribution of liquid precipitation. Even though 2019 had less rain occurrence (Pq) than 2020, the 2019 rain events lasted longer than in 2020, with nearly identical values of cumulated precipitation (Pc) for both: 275.21 mm in 2019 and 250.21 mm in 2020. The 2021 growing season had higher Pc despite the small amount of liquid precipitation per rain (<5 mm), probably because of the additional 18 days included, which encompassed 32 rain events across the 2021 growing season.

**Table 3.** Growing season meteorological standards from plantation to ICP.

| Growing Season | Days until ICP | Pmax | Pc | Pq | Tmax | Tmin | Tave | SRmax |
|---|---|---|---|---|---|---|---|---|
| | (Days) | (mm) | (mm) | (Events) | (°C) | (°C) | (°C) | (W/m$^2$) |
| 2019 | 80 | 22.81 | 275.21 | 31 | 30.77 | −1.10 | 15.27 | 1276.90 |
| 2020 | 80 | 23.21 | 250.21 | 22 | 34.33 | 4.10 | 20.31 | 1276.90 |
| 2021 | 98 | 4.80 | 405.00 | 32 | 32.64 | −1.18 | 17.58 | 1279.00 |

Unlike the liquid precipitation behavior, the temperature patterns were similar for the 2019 and 2021 growing seasons in the form of Tmax, Tmin, and Tave (Tave < 18 °C). The 2020 growing season had the highest Tmax (34.33 °C) and the lowest temperature range. Regarding solar radiation, all growing seasons had an SRmax near 1280 W/m$^2$. The difference between them is the number of times the daily SR obtained values close to SRmax. This happened more often during the 2020 growing season. These data help to explain why this season was the warmest, having T ≥ 30 °C more often than the 2019 and 2021 growing seasons.

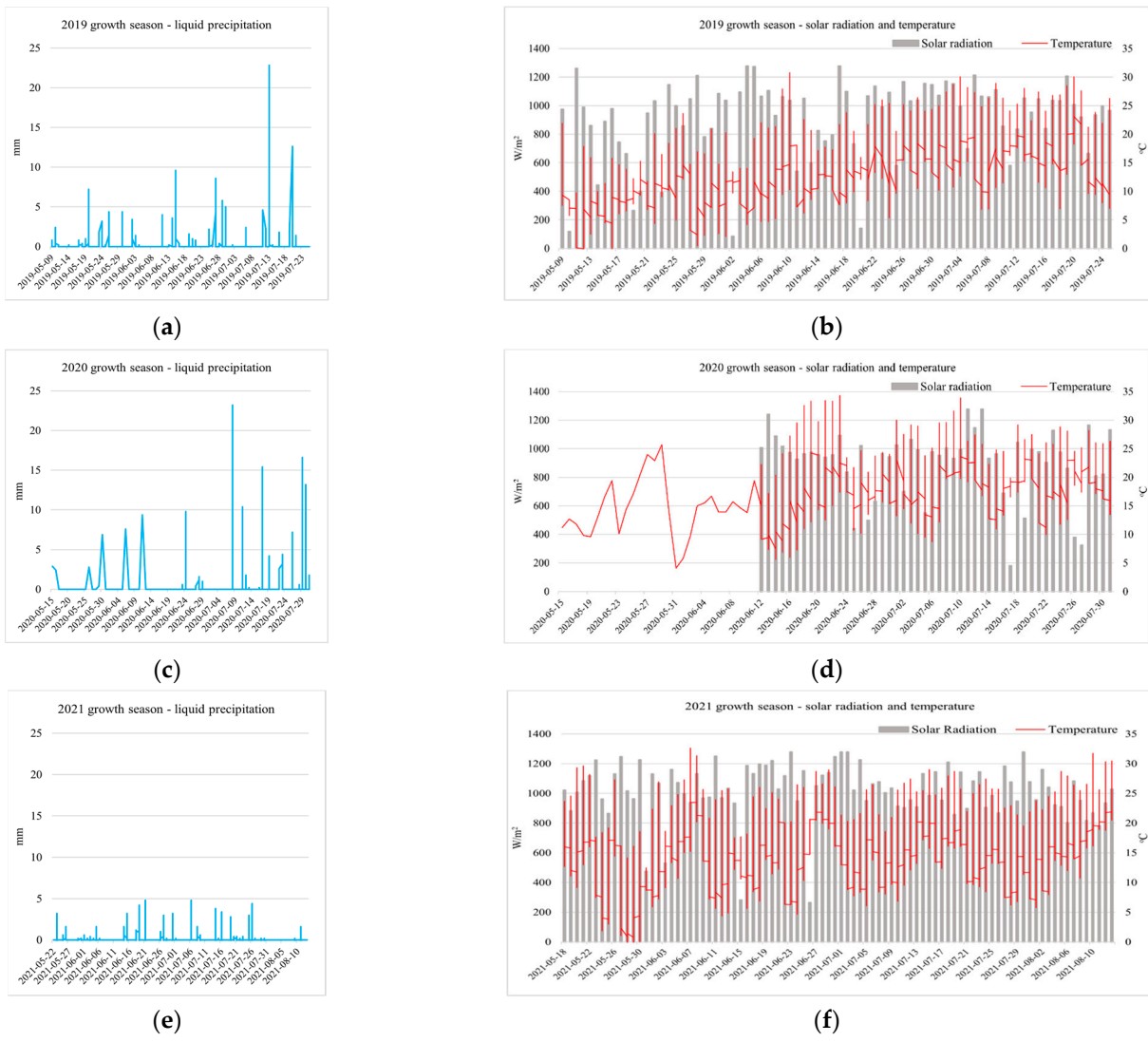

**Figure 7.** Meteorological critical data for all growing seasons, specifically precipitation, solar radiation, and temperature. The graphics concern the period until the achievement of growth curve stabilization, which means that the period covered is variable according to the growing season. 2019 growing season: 2019/05/09–2019/07/25 (**a**) liquid precipitation (mm) and (**b**) solar radiation (W/m$^2$) and temperature (°C); 2020 growing season: 2020/05/15–2020/07/31 (**c**) liquid precipitation (mm), and (**d**) solar radiation (W/m$^2$) and temperature (°C); and 2021 growing season: 2021/05/18–2021/08/21, (**e**) liquid precipitation (mm) and (**f**) solar radiation (W/m$^2$) and temperature (°C).

## 4. Discussion

The results indicate that RD values are almost invariable over the field and even among growing seasons. This result is not surprising, since root development is a physiological issue [14,15,24,25,27]. Although RD does not seem sensitive to meteorological and infield variations, the stabilization momentum should still vary across growing seasons due to plant-physiological development. This behavior is compatible with the results of some studies that indicate the influence of water on root density and potato plant development [11,12,23,25–27].

In contrast to RD results, H data demonstrate that plant height is sensitive to spatial variations, either in-field (MZs) or field placement (growing season), and to meteorological variation. This result is compatible with other research stating that these factors are important constraints on plant development [10,14,15,20–23,26]. Owing to meteorological

standards and spatial variation, the H stabilization momentum and average values vary over growing seasons when results are compared.

The inter-MZ absolute values for H and the H growth curve stabilization momentum should vary over growing seasons. Together, meteorological data and H indicate that the magnitude of these variations is related to weather patterns and local or regional characteristics. Nevertheless, the heterogeneity of plant H values across seasons does not affect the shape of the growth curve and its related H rate, indicating its independence from environmental fluctuations. Thus, the maximum H value should be reached at the same time for all crop plants, independent of in-field variations, but the period until the ICP is reached will vary across growing seasons. Furthermore, no standard values of plant height were identified as key values to predict the momentum of H stabilization, probably because they are related to the physiological constraints of the crop.

Regarding results, it is possible to affirm that, whatever the fluctuation observed in H values, the growth curve pattern may be replicable. Thus, the H growth rate was appropriate for describing plant growth dynamics, since it relates to the changing intensity in H increase over time, independently from the magnitude or the range of values reached. Using the H growth rate instead of the H value itself improved the method's performance and applicability as the gradient normalizes the data.

Corroborating what some authors found [20,22,24,26,48], a strong correlation between H and RD growth dynamics was detected. Moreover, the results indicate that it is independent of placement, irrigation method, or weather variations. The RD values are almost homogeneous, while the H values are variable according to the field MZs. This behavior is expected as the atmosphere is a less stable environment than the soil [15,26]. Probably, the growth dynamics of the plants change according to the environmental variations to supply for in-soil development [15,20–22,24,25,27].

As for the regional weather pattern of the Quebec agriculture region, temperature, precipitation, and solar radiation were improved mid-season (October) [9,41,49]. Similarly, the length of the growing season can change according to these conditions [8–10]. For example, during the 2021 growing season, plants needed more days until stabilization, probably because of the amount and distribution of net liquid precipitation. Despite the strong influences of climate on the growth of plants, especially on the H value, the growth curve stabilization occurs when RD reaches its maximum value. This behavior is probably the natural response of the crop for ensuring its physiological pattern, as already described in the literature [8–10,20–22].

In general, the results also indicate that variations in meteorological standards should impact crop yield because plant development patterns may shift in time, and the duration of the growing season varies. Instead, the plant growing dynamic is not reshaped, but the length of phenological stages change relatively to plants growth patterns as it is described in the literature [8–10,20–22,27,50]. This plant adaptation to meteorological pattern changes may cause crop yield constraints.

Because the H growth rate is independent of spatial and meteorological variations, it succeeds in delivering both the H and the RD stabilization momentum (H rate $\approx 0$). The results prove that it was possible to determine the ICP for all growing seasons studied, independent of meteorological or spatial variations. Thus, the H pattern is a key to accessing the crop development dynamics via remote sensing data. Similarly, the stability of the obtained results leads us to consider that, at an operational level, using the H growth rate instead of real H values should improve the applicability of this methodology since, no matter the range of H values, the growth rate may indicate the beginning of tuberization.

Even though this approach has performed well for Russet Burbank cultivated in the fields studied, we consider that more data from other regions and potato types may increase the reach and application of this technical approach.

## 5. Conclusions

Meteorological and infield variations have a critical impact on plant height (H) but not on root depth (RD) development, which is almost invariable across the field. However, H growth rate demonstrated its potential for describing plant development dynamics and, consequently, root stabilization momentum. Likewise, the plant growth rate is the key to identifying the beginning of the tuberization period, and therefore the ICP in potato crop fields.

Regardless, the changes in meteorological standards over the seasons highly impact the plant development dynamics. Hence, the plant development phases can shift in time, and the growing season duration should vary. Instead, the plant growing dynamic is not reshaped, which impacts crop yield. For example, in an extreme situation, depending on the weather, immature tubers can be harvested at the end of the growing season.

This research has proven that this critical momentum for potato crop fields, the ICP, can be detected via the proposed remote-sensing approach without any punctual invasive sampling. Moreover, this approach is based exclusively on plant development dynamics and does not consider predefined values, coefficients, or dates. In the same way, it allows for spatial and temporal flexibility. These characteristics ensure the ability to apply this approach regardless of field placement, irrigation method, or soil characteristics.

Knowing the ICP with precision may have several positive effects on all production processes, as it is crucial in tuber development. Since the proposed approach should accurately determine when to start irrigation, it may avoid damages caused by early irrigation and reduce water consumption. In addition, it is possible to use this approach to benefit other practices in precision agriculture and smart farming.

To our knowledge, this is the first study aiming to develop an approach exclusively based on the plant growth dynamics and directly using plant height measurements from UAV MDEs. More data from different fields areas and from different potato types to increase the reach of this technical approach are currently being acquired.

**Author Contributions:** Conceptualization, S.M.; methodology, S.M. and R.L.; validation, S.M.; formal analysis, S.M.; investigation, S.M.; resources, A.C.; data curation, S.M. and R.L.; writing—original draft preparation, S.M.; writing—review and editing, S.M.; supervision, S.M., K.C. and A.C.; project administration, K.C.; funding acquisition, K.C. All authors have read and agreed to the published version of the manuscript.

**Funding:** This research was funded by Mitacs Accélération and Consortium de recherche sur la pomme de terre du Québec (CRPTQ) grant number [IT18419].

**Institutional Review Board Statement:** Not applicable.

**Informed Consent Statement:** Not applicable.

**Data Availability Statement:** Not applicable.

**Acknowledgments:** This research is supported by the Institut National de la Recherche Scientifique (INRS—Centre eau terre environnement), Agriculture and Agri-Food Canada, Maxi-Pant, Mitacs, and Consortium de recherche sur la pomme de terre du Québec. We thank all of you for your financial, technical, and practical assistance.

**Conflicts of Interest:** The authors declare no conflict of interest.

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
