# Peer review of "Determining the Beginning of Potato Tuberization Period Using Plant Height Detected by Drone for Irrigation Purposes"

_agronomy, doi:10.3390/agronomy13020492_

Round 1

Reviewer 1 Report (Previous Reviewer 3)

Dear author, thank you very much for opportunity in review your manuscript. I consider the authors made important changes in the manuscript and it was highly improved. I recommend the publication of the manuscript in its current form. Best regards,

Author Response

Thank you for your recommendation.

Best regards,

Reviewer 2 Report (Previous Reviewer 2)

My opinion is that the authors have accepted all the reviewers' suggestions and that the manuscript is now correct and can be published in the presented form.

Author Response

Thank you for your recommendation.

Best regards,

Reviewer 3 Report (New Reviewer)

1.The small sample size and the difference in  treatments of the 3-year tests will lead to  differences in results and the decrease in reliability。

2.The title of the article does not correspond to the results.

3.The paper does not make clear the relationship between ICP and RD, H, meteorology, which needs to be emphasized and supported by data, resulting in the overall incompleteness of the paper

4.It is recommended to cite the latest published literature in the introduction.

5.The discussion and conclusion need to be further clarified and refined.

6.UAVs data can be used to acquire for more VIs, Vegetation index can help determine ICP better. So we suggest this paper can consider VIs to improve its novelty.

Author Response

I would like to thank you for your comments. They made me improve my research.
Please see the attachment. 

This manuscript is a resubmission of an earlier submission. The following is a list of the peer review reports and author responses from that submission.

Round 1

Reviewer 1 Report

This manuscript attempts to study the Critical point for potato crop irrigation by UAV technology. However, there are too many defects in it, which is more like a work report than an academic paper.

Author Response

Owing to the lack of precision in your Revision Report, we are attaching the corrections demanded by the other reviewers. I hope the changes made in the document will address your expectations.

Please, see the attached file, and feel free to send me more comments.

Reviewer 2 Report

My impression is that the manuscript entitled "Critical point for potato crop irrigation guided by UAV remote sensing" represents an interesting and innovative approach to the research of this topic, but some corrections would have to be made in order for it to be accepted for publishing.

The points listed below should be considered or revised:

Line 17: After the text “irrigation critical point”, you should add (ICP), as to define the abbreviation used in the abstract.

 Line 18; Line 206; Figure 6; etc:   It is common to express the correlation as a coefficient of correlation (Pearson or Spearman) or as a coefficient of determination (possible in %). In the text of the manuscript, it should be clearly pointed out which parameters and values are in question. Also, it is common to test the statistical significance of the correlation, which has not been done in this manuscript.

In the introduction, it stands: The main potato physiology characteristics used in this research are: (i) tuber maturity and quality depend on soil moist and nutrients. However, no additional data on these parameters is presented in further text.

I suggest explaining the goals and scientific impact of the research in more detail. The link between potato crop growth dynamic, depth of root system and the beginning of tuberization period is well-established and known, and already present in various literature, as well as the possibilities of plant height determination using photogrammetric techniques.

"This momentum is called here the irrigation critical point (ICP) because water supply becomes a farmer’s concern since the tuber development stage begins." It should be clearly expressed that “Critical point for irrigation” refers only to one of the critical phases in potato plant development (tubers development – the beginning of tuberization period). The role and significance of irrigation in other critical phases of potato plant development that precede the beginning of tuberization period should not be understated. Furthermore, it should be stated that, despite the use of this method, the need for invasive soil sample taking remains - eg. in determining the soil moisture, content of nutrients, fertirrigation demands etc.

Materials and Methods:

What area did each of the three potato crop fields cover?

Any information about the land cultivation on the three potato crop fields during three growth seasons would be useful (because of the root depth).

Please state was the preciseness of plant heighth measurement by aerial photography defined – and if yes, what is it?

On which sample (how many plants) was the root depth measured?

Line 57: Instead of: nutriments, it should stand: nutrients

Line 125: Instead of: Metrological dada, should stand: Meteorological data

Figure 7(c). Vertical axis title is incorect

Line 294: poxy of plant growth stabilization. Please correct

In the chapter Discussion, you should add views and opinions of other authos on this topic.

The conclusions are written in too much detail. Some paragraphs from the Conclusion could/should be moved to Discussion. 

Some statements in the Conclusion are not adequately argumented and backed-up with data in the Results. Eg. "Therefore, this approach may increase the crop 310 yield, reduce production costs with fertilizers and pesticides, as well as improve water quality and reduce its consumption.", etc.

I propose a medium revision, giving the possibility to the authors to correct the manuscript and address the comments above.

Author Response

Please see the attached file, and feel free to send me more comments.

Reviewer 3 Report

The manuscript " Critical point for potato crop irrigation guided by UAV remote sensing" (agronomy-2036816), demonstrated the potential use UAVs to monitoring a water suppling demand in potato culture. The results showed interesting data based on remote sensing for irrigation system decisions.

The authors have done a large amount of work, but few references and critical analysis based on a scientific method and structure it’s not correct.

I had a slight impression that this work is linked to a doctoral research, which, however, was briefly written for publication. My concept this a maybe a the “half-manuscript”?!!

However, the manuscript has a merit to publication, and after major correction, specially a discussion topic, which doesn’t adequate and need major corrections. For example, where are references?

Minor correction in points its necessary by adjusting in English synthases.

However, in my conception, the manuscript is suitable for publication, after major corrections in discussion topic. For example, your conclusion its bigger than the discussion!

Points:

#1: There is a scope for improvement in the introduction section: a) scientific and economic contribution of the paper; b) prospectively to other plants to agronomic interest. Maybe a one paragraphs with potential economic by important this manuscript to science crop;

#2: Please. All standardization of nomenclature equipment/software when necessary. Example: Fabricant, City, State, Country (three-letter). Check all manuscript.

#3. Alphabetic order keywords;

#4. Check citation in manuscript, and all references following “Author Instructions” in Agronomy journal.

#5. Figures and tables need major correction, in specially to y-axis (all figures, check please!).

#6. Check all references, Its a not correct! years, authors, pages.

Minor points

L09. …plant growth, development….

Abstract. Writte your main objective; and importance this study.

L20. What is ICP?

151. 60 m

L160. dot

Figure 3. y-axis; its unit, but what is this? “elevation, height

L191. space

L222. 275,21, this correct? or 275.21?

L227. space

Table 2. (oC), this correct unit?

Figure 7. y-axis; its unit, but what is this?

Best regards

Author Response

(The authors gave the same response as above.)

Round 2

Reviewer 1 Report

I still haven't seen any substantial improvement in the manuscript. The main problems are as follows:
(1) The emphasis of this paper is to use remote sensing technology to carry out crop height, soil moisture and other related parameters, and then judge the irrigation critical point (ICP). There are many researches on these aspects, which is not a new field. I didn't see the author review the existing researches, find the problems, and carry out targeted research, but only work according to his own imagination.
(2) The method is not the process of field work, but the idea or scheme to solve problems.
(3) In the part of results, you only analyzed the relationship between root depth and plant height, and the relationship between height and growth curve. We didn't see any inherent inevitable relationship among UAV, altitude, ICP, etc.
(4) I haven't seen any instructive conclusions or specific problems solved in the manuscript.
The overall feeling is that the content of scientific research is not enough, the research points that need to be broken through are not captured, the research plan is simple, and the analysis and mining are not enough.
In the future research, it is suggested to improve the value of the manuscript from the following methods:
(1) Systematically analyze the existing research methods in this field and the difficulties that need to be broken through;
(2) Look for innovations in methods, or dig meaningful content from applications.

Author Response

Dear reviewer,

I'll try to answer your concerns above:

  1. The emphasis of this paper is to use remote sensing technology to carry out crop height and root depth relationship. This is the potato crop stage named here the ICP. We’ve never mentioned  soil moisture. We are proposing an approach completely based on the plant growth dynamic to meet precisely the tuberization period beginning. In this concern, we’ve cared out many research about potato crop physiology, photogrammetry, and son on, but no existing research in the main subject of this research was found.
  2. We have 3 years of field works, weekly during the whole growth season. And, at the field we were able to better understand the results that we’ve had.
  3. It’s because the main idea here it to verify the correlation between root depth and plant height to infer about using the growth curve stabilization, and better the growth rate, to meet the tuberization period beginning.
  4. We are proposing to drone images instead of old plantation calendar method to determine when the irrigation period may start. This is the difficult that need tome broken: late or early irrigation determination because of the old methods to determine when the tuberization period starts.
  5. We’re presenting an innovation (see the last point).

I sincerely appreciate your points, and I hope my answers are good enough to better clarify your concerns. 

Reviewer 2 Report

Journal: Agronomy

 Manuscript Number: agronomy-2036816 - Revised Version Review Request

 Title: Critical point for potato crop irrigation guided by UAV remote sensing

Authors:  Sarah Martins*, Rachid Lhissou, Karem Chokmani, Athyna N. Cambouris

I thank the authors for accepting the reviewer's suggestions. The new version of manuscript: "Critical point for potato crop irrigation guided by UAV remote sensing" is acceptable and can be published as a Original Research Article in Agronomy journal.

Author Response

Thank you for your revision, it was very important for my work improvement.

Reviewer 3 Report

I consider the authors made important changes in the manuscript and it was highly improved. I would like to thank the authors for addressing my comments. I recommend the publication of the manuscript in its current form. Best regards,

Author Response

(The authors gave the same response as above.)
